# Artificial Neural Network Prediction of Compliance Coefficients for Composite Shear Keys of Built-Up Timber Beams

**DOI:** 10.3390/ma17133246

**Published:** 2024-07-02

**Authors:** Irene A. Ladnykh, Nabi Ibadov, Hubert Anysz

**Affiliations:** Faculty of Civil Engineering, Warsaw University of Technology, 00-661 Warsaw, Poland; irene.ladnykh.dokt@pw.edu.pl (I.A.L.); nabi.ibadov@pw.edu.pl (N.I.)

**Keywords:** built-up timber beam, composite materials, keyed timber beam, machine learning, mechanical malleable bonds, strengthening, timber structure

## Abstract

This article explores the possibility of predicting the compliance coefficients for composite shear keys of built-up timber beams using artificial neural networks. The compliance coefficients determine the stresses and deflections of built-up timber beams. The article analyzes current theoretical methods for designing wooden built-up timber beams with shear keys and possible ways of applying them in modern construction. One of the design methods, based on the use of the compliance coefficients, is also discussed in detail. The novelty of this research is that the authors of the article collected, analysed, and combined data on the experimental values of the compliance coefficient for composite shear keys of built-up timber beams obtained by different researchers and published in other studies. For the first time, the authors of this article generated a table of input and output data for predicting compliance coefficients based on the analysis of the literature and collected data by the authors. As a result of this research, the article’s authors proposed an artificial neural network (ANN) architecture and determined the mean absolute percentage error for the compliance coefficients k_w_ and k_i_, which are equal to 0.054% and 0.052%, respectively. The proposed architecture can be used for practical application in designing built-up timber beams using various composite shear keys.

## 1. Introduction

Timber structural members can be solid, glued laminated timber (synonym: glulam)*,* laminated veneer lumber [1,2,3], timber–concrete composite beams [4,5,6,7], and built-up timber beams without glue (synonym keyed timber beams) [8,9]. A keyed timber beam consists of individual parts of wood, which are fastened along the height of the cross-section using a shear key (synonym: mechanical malleable bonds). Shear keys are nails, bolts, plates, oak keys, dowels, and holders made of composite material with no glue (Figure 1) [8,9,10,11,12]. A keyed timber beam is used for generating a larger composite bending section with shear keys. Shear key or mechanical malleable bonds are mainly used to prevent slip between layers. For example, a wooden Derevyagin beam consists of two or more wooden layers or details interconnected using shear keys from oak dowels [8,9].

Built-up timber beams are an alternative to the large solid-sawn timber cross-sections and glulam. Nowadays, glulam is used for long-span structures instead of keyed timber beams. However, the researchers [10,11,12,13] proved experimentally that the glulam with through-beam cracks has the same mechanical characteristics as the keyed beam without connection. They suggested using composite mechanical malleable bonds to strengthen the stiffness and strength of the glulam beams instead of classical methods of strengthening, such as:Reinforcing clamps made of metal (Figure 2a) [12].Reinforcing elements (such as metal bolts) [12].Reinforcement glued rods [10,11].Strengthening of timber structures stressed mainly by bending or shear using externally bonded FRP materials (using carbon, glass, and aramid strips and rods) placed on the surface of strengthened elements [14,15,16,17,18,19,20,21,22,23,24,25].Gluing carbon strip and rod inserted into groove placed on the surface of strengthened element [15,16,17].Reinforcing mechanical malleable bonds in the form of closed and non-closed contours of composite material along the length of the cross sections (Figure 2b) [13,25].

For reinforcement glulam with through-beam cracks, authors in their papers [13,25] proposed using mechanical malleable bonds in closed and non-closed contours made of composite materials, such as unidirectional carbon fiber tapes and fiberglass. The authors proved that glulam with through-beam cracks is reinforced with composite mechanical malleable bonds, which works like a keyed timber beam. Their design model of the keyed timber beam is based on using the compliance coefficients. There are different methods to design the keyed timber beam. Still, all methods based on experimental data and the strengthening of the other mechanical malleable bonds depend mostly on the mechanical characteristics of the mechanical malleable bond material. Every new type of mechanical malleable bond should have been tested and have obtained new experimental data. All these factors negatively affected the wide application keyed timber beam with mechanical malleable bonds made of different composite materials.

This research paper is a way to predict the compliance coefficients for the design of built-up beams using artificial neural networks. The authors of this article collected, analysed, and combined data on the experimental values of the compliance coefficient for composite shear keys of built-up timber beams obtained by different researchers and published in other studies. Based on these collected data, for the first time, authors generated a table of input and output data for predicting compliance coefficients. The novelty of this research is the architecture of a neural network, which can be fed with new data on other types of composite mechanical malleable bonds and their compliance coefficients in the future. For the first time, the developed neural network will allow other researchers and engineers to estimate compliance coefficients for the mechanical malleable bonds made of composite materials based on data on the type of connection, the failed load for the two-cut sample, the reinforcement coefficient corresponding to the failed load, the number of wooden layers in a timber element, the reinforcement coefficient for a wooden element, and the length of the timber element. In its current form, it can be a tool for a rough assessment of the compliance coefficients. In our future work, we plan to compare methods for calculating keyed timber beams and develop a method for predicting various parameters necessary for calculating keyed timber beams using different methods.

## 2. Materials and Methods

### 2.1. Overview of Built-Up Timber Beam with Mechanical Malleable Bonds Made of Composite Materials Design

The theoretical model of a built-up beam is described in [8,9]. Keyed timber beams are always compared with solid timber beams and keyed timber beams without being connected.

When a solid beam, of depth d, is loaded to produce a positive moment (the moment that causes the top side of the beam to go into compression, while the bottom side is tensioned), the beam deflects downwards. The top side shortens as the bottom side lengthens. At the centroid (mid-height for a rectangular section), commonly called the neutral axis of a beam, the beam does not change length when subjected to small transverse deflections [8]. See Figure 3 [8].

When two beams, each of depth $\frac{d}{2}$, are stacked atop each other without being connected (no interaction between the two beams) and loaded to produce a positive moment, each beam has both compression and tension components. The shortening of the top face of the bottom member, combined with the lengthening of the bottom face of the top member, results in substantial slippage between these two layers [8,9]. See Figure 3 [8].

The stiffness and strength of a keyed timber beam are between the stiffness and strength of a solid timber beam and the stiffness and strength of a keyed timber beam without being connected. The stiffness of mechanical malleable bonds determines the stiffness and strength of the keyed timber beam.

There are several ways to design the built-up beam. The way to design depends on the type of shear keys or mechanical malleable bonds and the national codes and practices that have been designed. For instance, the Eurocode [3] uses a design approximation based on an effective stiffness “EI” (modulus of elasticity multiplied by the moment of inertia), which is numerically convenient so long as the stiffness of the shear connectors is known.

One might consider another approach to design the built-up beam, which was suggested by Karlsen using adjustment factors to modify the moment of inertia. [8,9,26] This method is used to design the keyed timber beam composed, which was published in [13,25]. This method uses the compliance coefficient $k_{w}$ and $k_{i}$ to determine the critical bending stress and final deflection of the keyed timber beam. These compliance coefficients, $k_{w}$ and $k_{i}$, depend on the type and material of the mechanical malleable bonds. The compliance coefficient $k_{w}$ is applied as a reduction coefficient to the moment of resistance when determining the critical bending stress of the built-up timber beam. The compliance coefficient $k_{i}$ is applied as a reduction coefficient to the moment of inertia when determining the critical deflection of the built-up timber beam.

Building codes [26] contain the tables with this coefficient for mechanical malleable bonds such as nails, bolts, plates, oak keys, and dowels. Previously, it was noted that there are two modern types of composite mechanical malleable bonds, which are presented in the research paper [13,25]. The composite mechanical malleable bonds are the composite material, such as unidirectional carbon fiber tapes or fiberglass, which is glued on a dry wooden surface with epoxy adhesive in the form of the closed and non-closed contours along the height of the cross-section of the beam. The composite mechanical malleable bonds look like clamping or straps. It is denoted that the mechanical malleable bond type I is made of unidirectional carbon fiber tapes on an epoxy adhesive (Figure 4a) [13] and the mechanical malleable bond type II is made of fiberglass on an epoxy adhesive (Figure 4b) [25].

At the same time, researchers note the possibility of using mechanical malleable bonds in timber structural elements in new construction and strengthening existing timber structures only made from pine tree. Figure 5 shows an example of reinforcement wooden arches using mechanical malleable bonds of Type I.

The main advantages of such mechanical malleable bonds are high corrosion resistance when strengthening a wooden element, their ability to stop the development of cracks, and the ease and convenience of installation. The load-bearing capacity and deformation of built-up timber beams with mechanical malleable bonds made of composite materials are calculated by reducing compliance coefficients for the moment of resistance when checking for strength$,\;k_{w}$, and reducing compliance coefficients for the moment of inertia when checking for deformability of the element$,\;k_{i}$. The researchers presented the values of compliance coefficients for specific composite materials [13,25].

The first way to determine the compliance coefficients is the experimental evaluation of the stiffness coefficient of the mechanical malleable bond according to the keyed timber beam theory by Pleshkov using a test setup, as presented in Figure 6. The research paper [13] also proposed to calculate and consider the reinforcement coefficient (K_arm_)—the ratio of the length of the tape (connection width) to the length of the cut (cutting lengths) (Figure 6). Then, using the stiffness coefficient and the keyed timber beam theory by Pleshkov, the coefficients of compliance $k_{w}$ and $k_{i}$ are determined.

There is another way to determine compliance coefficients $k_{w}$ and $k_{i}$. They can be obtained based on the following results of the experiment: the amount of the normal stresses of the built-up beam with mechanical malleable bonds, the normal stresses of the solid timber beam, and the amount of deflection for the built-up timber beam and solid timber beam. This process is presented in Figure 7.

Then, using Formulas (1) and (2) the coefficients of compliance $k_{w}$ and $k_{i}$ are determined according to [8,9,13,25].
(1)$$k_{w}=\frac{\sigma_{cal}}{\sigma_{exp}}$$
where $\sigma_{cal}$—calculation—designed stresses for the solid timber beam, MPa; $\sigma_{exp}$—experimental—designed stresses for the built-up timber beam, MPa.
(2)$$k_{i}=\frac{u_{cal}}{u_{exp}}$$
where $u_{cal}$—calculation—deflection for the solid timber beam, mm; $u_{exp}$—experimental—deflection for the built-up timber beam, mm.

The mechanical characteristics of the composite material (fiberglass and unidirectional carbon fiber tape), as well as adhesive and composite material, were also determined.

Independently from the described above methods (based on materials’ tests) of determining the coefficients on the subject, if these coefficients are determined for several conditions, it is possible to collect the results and build separate machine-learning models that are able to predict $k_{w}$ or $k_{i}$ with high accuracy. The flowchart of the proposed approach is presented in Figure 8.

### 2.2. The Collected Data of Compliance Coefficient

According to [13,25], the following factors that affect the mechanical malleable bond’s mechanical characteristics and coefficients $k_{w}$ and $k_{i}$ have been identified to determine the input characteristics:Mechanical characteristics of the composite material from which the mechanical malleable bonds are made, such as fiberglass and glue. This parameter considers the bond’s material itself (unidirectional carbon fiber tape or fiberglass), as well as the glue that is used to glue the connection to the wood surface.The reinforcement coefficient.The thickness of the glued composite material and epoxy tape after drying.The number of tape layers.Lengths of the built-up timber beam.The number of layers in the built-up timber beam.

The authors of this article collected data on compliance coefficients $k_{w}$ and $k_{i}$ for composite malleable bonds (presented in [13,25]) and combined them into Table 1, Table 2 and Table 3. According to [13,25], the compliance coefficients presented in Table 1, Table 2 and Table 3 depend on the following indicators:The type of malleable bond (I or II).The load at the elastic limit on the load-strain curve for the two-cut sample.The strain at the elastic limit on the load-strain curve for the two-cut sample.The reinforcement coefficient for two cut samples.The reinforcement coefficient for built-up timber elements.The number of bonded details by the height of the cross-section for built-up timber elements.The reinforcement coefficient and the length of the built-up timber element.Length of the built-up timber element.

Table 3 presents data for mechanical malleable bonds of Type II, which are made of fiberglass on an epoxy matrix.

The main limitation according to [13,25] is the possibility of using these compliance coefficients only for mechanical malleable bonds made from materials that the authors used in their research. To use other composite materials (fiberglass and unidirectional carbon fiber tape and glue) that differ from those presented in the research [13,25], it is necessary to reconduct research using one of the methods described above. This is the main problem of introducing mechanical malleable bonds as a closed and non-closed contour in the wide practice of construction and the use of various composite materials. Therefore, the authors of this study suggest using machine-learning methods to predict the coefficients $k_{w}$ and $k_{i}$ for composite mechanical malleable bonds made of the different types of composite materials and epoxy glue.

### 2.3. Application of Artificial Neural Network in Building Materials Research

Machine-learning algorithms are widely used in determining the mechanical properties and mix design of concrete including optimization [27,28,29,30,31,32,33,34,35,36,37], which made it possible to predict the compressive strength of concrete and cost reduction of laboratory tests. The research results show a high level of accuracy in predicting the mechanical characteristics of concrete, which makes it possible to introduce these methods into production.

In timber construction, machine learning is developing in the direction of visual quality control of wood [38] to evaluate the fire resistance of timber columns [39]. This is due to the peculiarities of wood as a building material, the size of the data sample, and the smaller spread of wooden construction in the world than concrete construction.

Machine-learning approaches can usually be divided into two main types: supervised machine learning and unsupervised machine learning [40,41]. The first of them is more often used to evaluate the mechanical properties of concrete, steel, and other construction materials [42,43,44]. Supervised machine-learning models consist of computer algorithms capable of generating patterns and hypotheses using the provided dataset to predict future values.

To solve the problem of compliance coefficient calculation, it is proposed to consider the possibility of using machine-learning algorithms to predict the compliance coefficients $k_{w}$ and $k_{i}$ based on the mechanical characteristics of the composite material.

## 3. Results

### 3.1. Essentials

This research presents the implementation of machine learning to the determination of the compliance coefficient for the design of the built-up beam with composite mechanical malleable bonds. Based on data presented in Table 1, Table 2 and Table 3 and experimental data from [13,25], artificial neural networks are built, which allow for the estimation of the compliance coefficients *k_w_* and *k_i_* for mechanical malleable bonds made of different types of composite materials. Unfortunately, data, which are presented on the building standards [26], cannot be used because they were determined for different types of mechanical malleable bond such as nails, bolts, plates, oak key, and dowels. The artificial neural networks estimate the compliance coefficients $k_{w}$ and $k_{i}$ based on the following inputs:the type of connection,the load at the elastic limit on the stress–strain curve for the two-cut sample,the strain at the elastic limit on the stress–strain curve for the two-cut sample,the reinforcement coefficient for two-cut samples,the reinforcement coefficient for built-up timber elements,the number of bonded details by the height of the cross-section for built-up timber elements,the length of the built-up timber element.

Machine learning is performed using the STATISTICA 13.1 software package (by Tibco). The practical application of this method is presented in Figure 9.

Based on the analysis of the factors that affect the mechanical characteristics of the technical malleable bonds, the following assumptions and limitations are introduced:

The adhesive consumption per 1 m^2^ of the surface is equal to the consumption specified by the manufacturer.

The adhesive on the surface of the wood is distributed evenly only in those areas where it is planned to install a mechanical malleable bond.

The installation of mechanical malleable bonds, as well as their subsequent drying, is conducted at the temperature and humidity specified by the manufacturer of the composite material and adhesive.

The drying time is also set by the manufacturer’s recommendations.

Only pine as a wood material is considered since researchers performed their tests only for this type of wood, as it is the most widely used.

If the stress–strain curve of the mechanical characteristics of the composite mechanical malleable bonds has sections with an inelastic nature of work, then in this case, we will take the stress and strain at the point of the limit of proportionality.

### 3.2. Creating a Database for Predicting the Compliance Coefficients $k_{w}$ and $k_{i}$

Database, which collected and created by the authors, consists of the values of the compliance coefficients $k_{w}$ and $k_{i}$ based on research [13,25] and Table 1 and Table 2 for the mechanical malleable bonds type I and Table 3 for the mechanical malleable bonds type II. The collected database consists of 123 rows (cases) for compliance coefficient $k_{w}$ and 125 for compliance coefficient $k_{i}$. The database is presented in Appendix A. Before the ANN is run all types of non-categorical inputs and the output are standardized with the linear-maximum method with the use of the following formula:(3)$$a_{i}^{\left(1\right)}=\frac{a_{i}^{\left(0\right)}}{max{\;a}^{\left(0\right)}}$$
where:$a_{i}^{\left(0\right)}$ is an i-th element of an input type a before standardization$\max a^{\left(0\right)}$ is the maximum value of an input type a$a_{i}^{\left(1\right)}$ is an i-th element of an input type a after standardization

To increase the reliability of the predictions, the cross-validation process is performed based on six folds (see Figure 10). Dividing the full dataset into three subsets (training, testing, and validating) is forced by the software. The artificial neural network is trained based on the training subset. The testing subset serves as a protection from overtraining the model. When the model is found, it is validated (tested) based on the validating subset. While dividing the dataset into subsets, special care is put into the proportional representation of cases in each subset. It refers to the inputs type where there are limited number cases, and all of them need to be represented in the training, testing, and validating dataset.

### 3.3. Results

The multilayer perceptron (MLP) type of ANN is chosen for the predictions. The Statistica 13.1 software limits the number of hidden layers to one. The results of predicting the compliance coefficients for timber composite elements on mechanical connections using machine-learning algorithms are presented in Table 4 and Table 5 for compliance coefficients $k_{w}$ and $k_{i}$, respectively. The architecture of the best ANNs found for each fold is described in these tables (e.g., MLP 6-11-1 means multilayer perceptron type of ANN with six inputs, 11 neurons in the hidden layer, and one output). Also, the activations functions are presented there. To assess the accuracy of predictions, MAPE (mean absolute percentage error) is applied. It is defined with the following formula:(4)$$MAPE=\frac{1}{N}\sum_{i=1}^{N}\left|\frac{y_{i}^{\prime }-y_{i}}{y_{i}}\right|\times 100\%$$
where:N is the number of cases in the subset$y_{i}^{\prime }$ is the i-th predicted value$y_{i}$ is the i-th observed value

The presented results in Table 4 show $k_{w}$ predictions, and in Table 5, the results of $k_{i}$ predictions are discussed in the following Section 4.

## 4. Discussion

As a result of the research, it is found that the compliance coefficients $k_{w}$ and $k_{i}$ can be predicted with artificial neural networks. Then, calculating the maximum destructive load of the mechanical malleable bonds is possible. The really low prediction error is achieved (averaged MAPE for six folds) for the compliance coefficient $k_{w}$, which is 0.054% for the validation subset. The averaged MAPE value for six folds for the compliance coefficient $k_{i}$ is 0.052% for the validation subset. Presented low values of MAPE indicate that the machine-learning model is accurate and can be used in real practice of designing wooden composite elements on mechanical malleable bonds.

In addition, it should be emphasised that MAPE is lower than the error of the instruments used to determine the coefficients in experiments. Therefore, the value of the predicted coefficient $k_{w}$ or $k_{i}$ MAPE will not affect the accuracy of the calculated values of the bearing capacity and deformability of wooden composite elements with mechanical connections in the form of the closed and non-closed contours of composite materials. To use the presented approach with the neural network, it is necessary to determine the input data shown in Figure 9. Special care was taken while the models were created. The step-up and step-down procedures are used to select the best set of inputs from the six available. It is confirmed that the lowest possible prediction errors are achieved only if all types of inputs are used. Any other combination of input types made the MAPE higher. Secondly, not to suggest (in the ANN training processes) which input influences the output the most, linear maximum standardization is applied. As the size of the database is rather low, the cross-validation is applied, so six separate ANN predicting models are found for $k_{w}$ predictions and another six models are built for $k_{i}$ predictions. The errors are averaged from these six folds separately for $k_{w}$ and $k_{i}$. All the above-mentioned subprocesses ensure that the result i.e., really low errors of prediction, are not achieved by chance, and that the prediction can be used for designing composite mechanical malleable bonds. Despite the really high accuracy of predictions, it has to be remembered that the method can be applied only for bonds:-Type 1 and Type 2 (described in earlier sections)-Joining the pinwood beams-Where inputs to ANN are within the existing ranges specified in Appendix A, Table A1 and Table A2 (e.g., max number of wooden layers is 10; max beam length is 6 m)

Not keeping the limits may lead to a significant error increase.

The main scientific novelty of this study is the prediction of compliance coefficients for timber composite elements on mechanical malleable bonds, using machine-learning algorithms based on neural networks. This research describes the application of a machine-learning algorithm based on artificial neural networks for predicting compliance coefficients for composite materials used for the design of a built-up timber beam. Experimental data on mechanical properties and compliance coefficients for mechanical malleable bonds are taken from published research by various authors. To ensure that the low errors are not achieved by a chance, the cross-validation procedure is applied. The creation of the machine-learning models was conducted in the STATISTICA 13.1 software. The MAPE type of error is chosen for assessing the models found. The machine-learning algorithm using artificial neural networks shows a high accuracy of created models. Low values of MAPE confirm that.

In conclusion, this study was based on the application of supervised machine-learning algorithms to predict compliance coefficients for timber composite elements. The algorithm used in this study shows a strong relationship between the actual and predicted output. The importance of these approaches in construction is evidenced by their high level of accuracy among real and predicted results. Supervised machine-learning approaches are gaining more and more popularity because their application provides results with high accuracy and minimizes the physical approach to practical work and the project’s total cost. In addition, input data, the type of material used, the mechanical characteristics of composite materials, and other parameters can be changed or added to study and compare the results of various machine-learning algorithms. The primary innovation of this work is the development of a neural network that enables the prediction of compliance coefficients for composite materials used in the design of built-up timber beams based on the basic mechanical properties of the joints without the need for additional testing of elements using these joints. This approach will reduce testing costs, minimize waste, and allow for the application of other types of composite materials in joint manufacturing.

## Figures and Tables

**Figure 1 materials-17-03246-f001:**
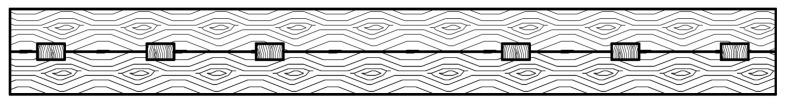
Two-layer built-up timber beam with mechanical malleable bonds [8].

**Figure 2 materials-17-03246-f002:**
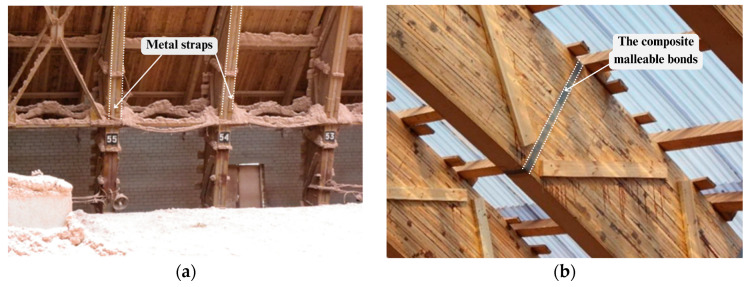
Wooden elements with reinforcements (own photo): (**a**) Wooden element reinforced with metal straps; (**b**) Wooden element reinforced with mechanical malleable bonds made of composite materials.

**Figure 3 materials-17-03246-f003:**
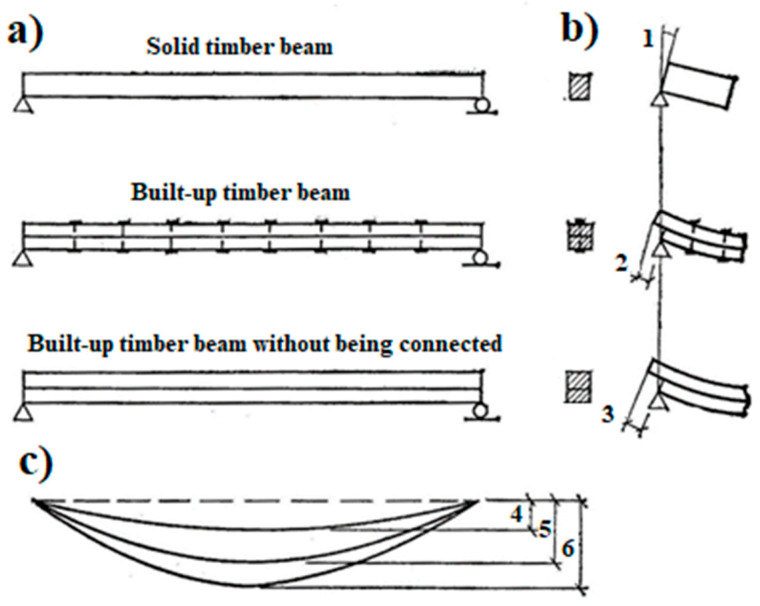
The types of timber beam [9]: (**a**) The different types of the beam; (**b**) the angle of rotation of the beam on the support (1—the rotation of the beam’s cross-section; 2—the interlayer slip between the top and the bottom layer of the built-up timber beam; 3—the interlayer slip between the top and the bottom layer of the solid timber beam without being connected); (**c**) the deflection for different types of the beams (4—the deflection for solid timber beam; 5—the deflection for built-up timber beam; 6—the deflection for built-up timber beam without being connected).

**Figure 4 materials-17-03246-f004:**
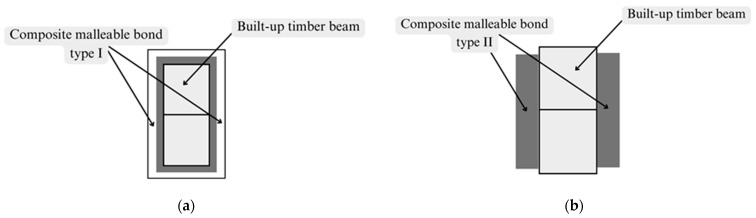
Mechanical malleable bonds based on composite materials (own elaboration figure): (**a**) Mechanical malleable bonds made of unidirectional carbon fiber tape on an epoxy matrix. (**b**) Mechanical malleable bonds made of fiberglass on an epoxy matrix.

**Figure 5 materials-17-03246-f005:**
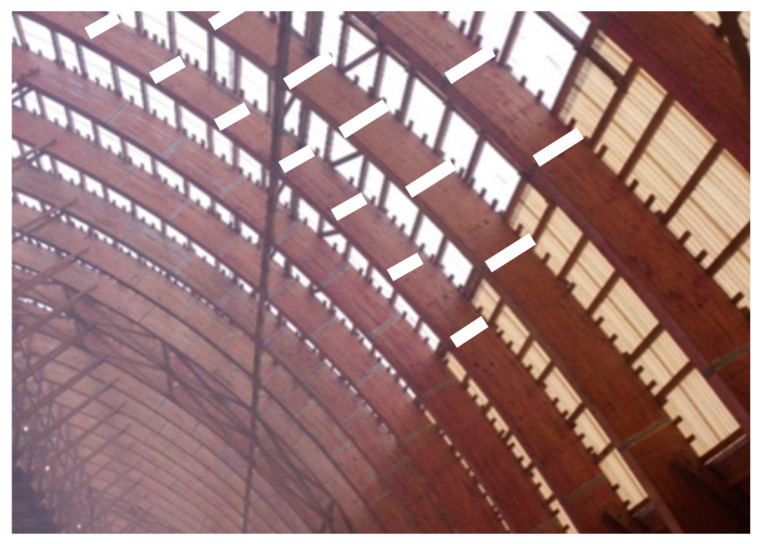
An example of strengthening timber arches using mechanical malleable bonds made of composite material (own photo). The white stripes are malleable mechanical bonds made of composite material [13].

**Figure 6 materials-17-03246-f006:**
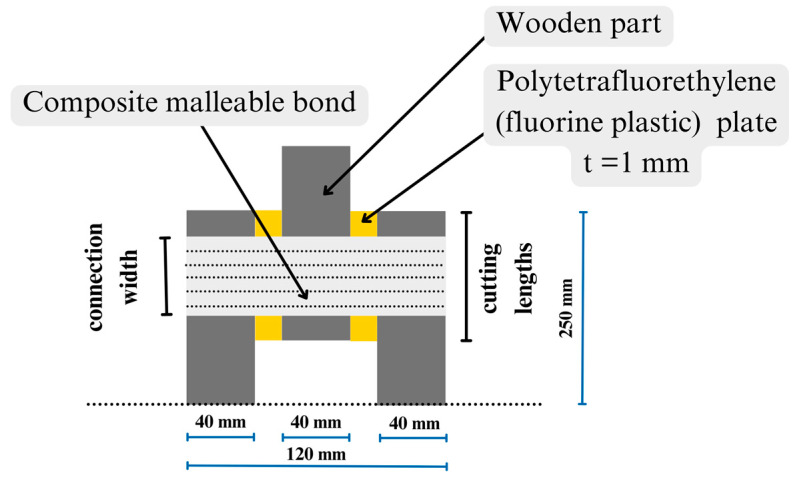
A two-cut sample for determining the mechanical characteristics of the mechanical malleable bonds for connecting built-up timber elements (own elaboration figure).

**Figure 7 materials-17-03246-f007:**
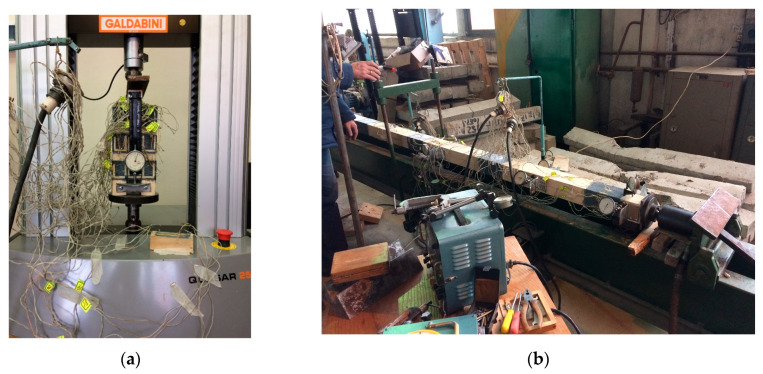
The process of determining the stiffness coefficient and the stress and deflection of built-up timber beam experimentally (own photo): (**a**) the testing process of a two-cut sample; (**b**) the testing process of a built-up timber beam.

**Figure 8 materials-17-03246-f008:**
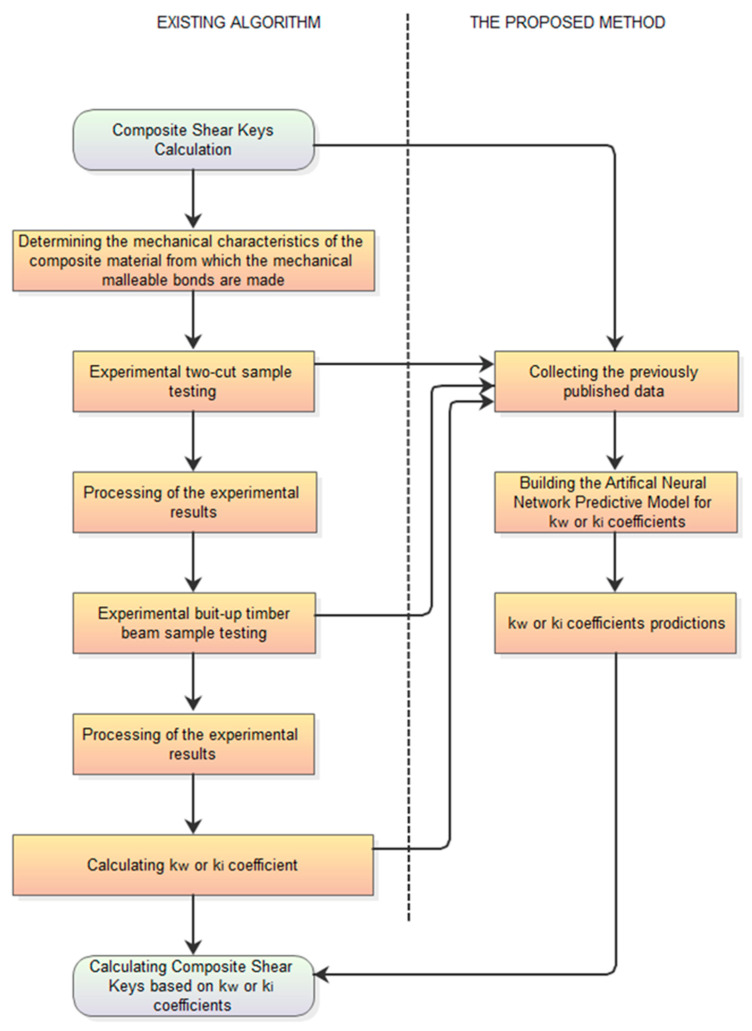
The proposed predictive approach to determination of $k_{w}$ or $k_{i}$ coefficients—the flowchart.

**Figure 9 materials-17-03246-f009:**
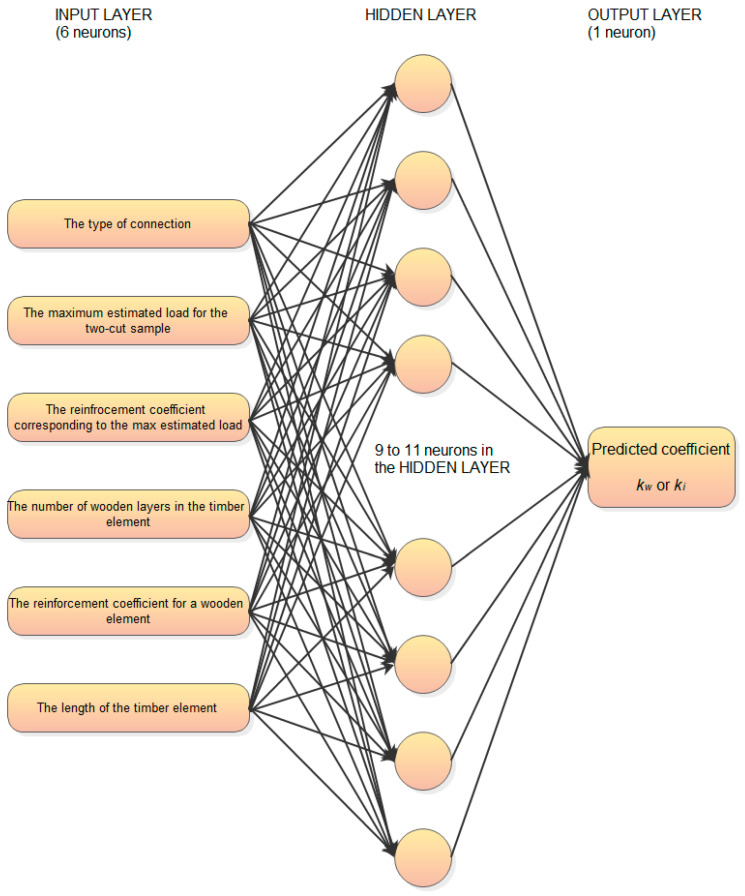
Block diagram of the practical application of machine learning in the prediction of compliance coefficients of the mechanical malleable bonds made of composite material (own elaboration figure).

**Figure 10 materials-17-03246-f010:**
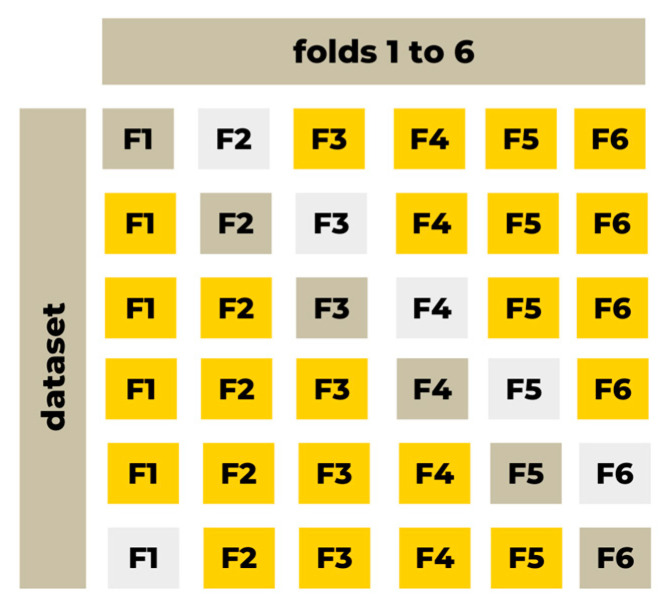
K-Fold cross-validation for machine learning (yellow—the training subset; light grey—the testing subset; grey—the validation subset) (own elaboration figure).

**Table 1 materials-17-03246-t001:** Results for coefficient$\;k_{w}$ for mechanical malleable bond type I [13].

Length of the Built-Up Timber Beam, m	Number of Layers in the Element	K_arm_ = 0.25	K_arm_ = 0.30	K_arm_ = 0.50	K_arm_ = 0.70	K_arm_ = 1.00
3	2	0.88	0.92	0.94	0.95	0.96
	3	0.78	0.80	0.82	0.84	0.86
	10	0.70	0.74	0.76	0.78	0.80
4	2	0.90	0.92	0.94	0.95	0.96
	3	0.84	0.86	0.88	0.90	0.92
	10	0.76	0.78	0.80	0.82	0.84
6 and more	2	0.92	0.93	0.94	0.95	0.96
	3	0.88	0.92	0.94	0.95	0.96
	10	0.80	0.82	0.84	0.86	0.88

**Table 2 materials-17-03246-t002:** Results for coefficient $k_{i}$ for mechanical malleable bond Type I [13].

Length of the Built-Up Timber Beam, m	Number of Layers in the Element	K_arm_ = 0.25	K_arm_ = 0.30	K_arm_ = 0.50	K_arm_ = 0.70	K_arm_ = 1.00
3	2	0.78	0.82	0.84	0.88	0.90
	3	0.52	0.55	0.56	0.59	0.60
	10	0.35	0.36	0.37	0.38	0.39
4	2	0.80	0.84	0.86	0.89	0.93
	3	0.55	0.58	0.60	0.62	0.64
	10	0.40	0.42	0.44	0.46	0.48
6 and more	2	0.81	0.85	0.87	0.89	0.93
	3	0.56	0.58	0.60	0.62	0.64
	10	0.41	0.42	0.46	0.46	0.48

**Table 3 materials-17-03246-t003:** Results for coefficients $k_{w}$ and $k_{i}$ for mechanical malleable bond Type II.

Length of the Built-Up Timber Element, m	$k_{w}$	$k_{i}$
1	0.90	0.85
3	0.95	0.90
6	0.95	0.90

**Table 4 materials-17-03246-t004:** Results for coefficient $k_{w}$ for the machine learning model.

	Fold 1	Fold 2	Fold 3	Fold 4	Fold 5	Fold 6	Mean Value
MAPE in % (training sample)	0.220%	0.148%	0.283%	0.147%	0.272%	0.068%	0.190%
MAPE in % (validation sample)	0.071%	0.047%	0.070%	0.046%	0.065%	0.027%	0.054%
MAPE in % (testing sample)	0.066%	0.060%	0.044%	0.047%	0.038%	0.027%	0.047%
Neural network architecture)	MLP 6-11-1	MLP 6-11-1	MLP 6-11-1	MLP 6-11-1	MLP 6-10-1	MLP 6-10-1	-
Activation function in the hidden layer	Tanh	Logistic	Logistic	Tanh	Tanh	Tanh	-
Activation function in output layer	Linear	Linear	Linear	Exponential	Exponential	Exponential	-

**Table 5 materials-17-03246-t005:** Results for coefficient $k_{i}$ for the machine-learning model.

	Fold 1	Fold 2	Fold 3	Fold 4	Fold 5	Fold 6	Mean Value
MAPE in % (training sample)	0.546%	0.426%	0.447%	0.251%	0.167%	0.197%	0.339%
MAPE in % (validation sample)	0.047%	0.071%	0.061%	0.062%	0.017%	0.052%	0.052%
MAPE in % (testing sample)	0.069%	0.112%	0.070%	0.074%	0.042%	0.042%	0.068%
Neural network architecture	MLP 6-11-1	MLP 6-10-1	MLP 6-9-1	MLP 6-9-1	MLP 6-10-1	MLP 6-11-1	-
Activation function in the hidden layer	Logistic	Logistic	Tanh	Tanh	Tanh	Tanh	-
Activation function in output layer	Linear	Linear	Linear	Tanh	Exponential	Exponential	-

## Data Availability

All data are referenced.

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
