# Peer review of "Artificial Neural Network Prediction of Compliance Coefficients for Composite Shear Keys of Built-Up Timber Beams"

_materials, 2024, doi:10.3390/ma17133246_

Round 1

Reviewer 1 Report (Previous Reviewer 1)

Comments and Suggestions for Authors

This paper discusses the possibility of using an artificial neural network to predict the flexibility coefficient of shear bonds of composite wooden beams. The theoretical method of designing timber beams with shear keys and the possible ways of its application in modern architecture are also analyzed. An input and output data table for predicting compliance coefficients is generated, and an artificial neural network structure is proposed. This paper has made some modifications based on the first draft, which has a certain reference value, but there are still several problems that need to be modified.

1. The last paragraph of the Introduction only briefly introduces the research content of this paper, but does not elaborate on the novelty of the previous relevant literature research, to reflect the value of this research, please revise;

2. On page 4, line 117, "Let's consider other ways..." These expressions are not correct, it is recommended to use the third-person expression, please check the full text, and modify it;

3. What is the meaning of the "•" in lines 258-266 on page 8? Why are they all represented by the same match, please explain.

4. The content of Figure 8 in the paper is too simple, and why is it represented in uppercase? It is suggested to redesign and draw it to increase readability and scientificity;

5. The appendix of the article is too long. It is suggested to use more concise data for representation, please modify it.

I hope the author can revise it carefully according to the requirements. Thank you.

Author Response

Dear Reviewer,

Please find the answers to your remarks in the attached file. All changes made to the article are marked in yellow, in the revised version of it.

Faithfully yours

The team of the Authors

Reviewer 2 Report (New Reviewer)

Comments and Suggestions for Authors

This work presents an artificial neural network that determines the compliance coefficients for Composite Shear Keys of Built-up Timber Beams based on a set of inputs.

Overall, the work is interesting; however, several points need to be addressed to demonstrate its contribution, reproducibility, and applicability.

  1. The literature contains many works that use the same approach (not the application) employed by the authors. These should be described and discussed to delineate the boundaries of current knowledge.
  2. In section 2.1, flowcharts should be added to complement the text, along with the equations that need to be considered at each step.
  3. The title of section 2 refers to neural networks, but the text generalizes with machine learning schemes.

a. While the concept of machine learning can be introduced, the majority should focus on neural networks.

b. Indeed, the choice of using neural networks needs to be justified.

c. Since this is a new application, other machine learning methods such as decision trees, support vector machines, k-nearest neighbors, etc., should be compared to demonstrate that the neural network provided better results.

d. There is no mention of the type of neural network used, activation functions, hidden layers, training algorithms, etc.

e. How is performance evaluated? Confusion matrices, accuracy, precision, F1-score, etc.?

  1. The diagram in Figure 8 needs to be supplemented or completely revised. It should show the structure of the neural network used: the number of inputs, hidden layers, output neurons, activation functions, etc.
  2. Was there one network for each coefficient or one for both?
  3. Was there data preprocessing? How were unbalanced data handled? How were data with different ranges managed? Was normalization performed?
  4. What are the results when considering a smaller number of inputs? Which inputs contribute the most? Which could be omitted?
  5. The use of k-fold = 6 needs to be justified. What happens if other values are used? For example, in the literature, it is common to use 5, which allows for an 80% training and 20% validation split. A higher number could lead to overfitting.
  6. The information in Tables 4 and 5 is unclear. Did the information in the last three rows change as the k-fold iterations progressed?
  7. The limitations of the work should be clearly stated.
  8. The contribution is not clear in either the introduction or the conclusion; it seems that only software was used on a database.
Comments on the Quality of English Language

 Minor editing of English language required

Author Response

Dear Reviewer,

Please find the answers to your remarks in the attached file. All changes made to the article are marked in yellow, in the revised version of it.

Faithfully yours

The team of the Authors

Reviewer 3 Report (New Reviewer)

Comments and Suggestions for Authors

Irene A. Ladnykh  et al proposed an Artificial Neural Network Prediction of Compliance Coefficients for Composite Shear Keys of Built-up Timber Beams. The authors collected and combined data on the experimental values of the compliance coefficient for composite shear keys of built-up timber beams obtained by different researchers and published in other studies.

I personally recommend that this manuscript should be published after some minor revisions:

1)      Please make the importance of the present work clearer.

2)      Rearrange the keywords so that they are in alphabetical order.

3)      Materials and Methods: which ISO did the authors use to choose the parameters for mechanical tests?

4)      Results: where is the elastic limit on the stress-strain curve? Can you represented used a machine learning algorithm? Why?

Author Response

Dear Reviewer,

Please find the answers to your remarks in the attached file. All changes made to the article are marked in yellow, in the revised version of it.

Faithfully yours

The team of the Authors

Round 2

Reviewer 2 Report (New Reviewer)

Comments and Suggestions for Authors

The authors have addressed all the comments and suggestions. This reviewer recommends the manuscript's acceptance.  

Comments on the Quality of English Language

 Minor editing of English language required

This manuscript is a resubmission of an earlier submission. The following is a list of the peer review reports and author responses from that submission.

Round 1

Reviewer 1 Report

Comments and Suggestions for Authors

This paper uses machine learning algorithm to predict the possibility of compliance coefficient of wood composite components on mechanical malleable keys made of composite materials, analyzes the application status of machine learning algorithm in the prediction of mechanical properties of building materials and wood structures, and gives an example of predicting the compliance coefficient of composite components based on data, and analyzes and discusses the calculated results. Thus, practical application recommendations can be determined. In this study, the method of machine learning is used to predict the compliance coefficient of wood, which has certain novelty. However, the following issues need to be revised before publication.

1. There are no quantifiable results in the abstract, which are basically qualitative conclusions, which cannot convince readers, please revise;

2. Please add references, copyright license and other information to the figure quoted in the paper, as shown in Figure 1,2,...... ;

3. As shown in Figure 7, the content is not clear enough and the font format is consistent with other graphics, it is suggested to modify it again.

4. Please indicate the meaning of different colors in Figure 8. In addition, it is suggested to combine Figure 7 and Figure 8 into one figure;

5. Please explain the meaning of Figure 9 in the paper, and what scientific meaning does it need to express?

6. The discussion part of this paper is too simple. Please re-write this part based on the research content and results of this paper to reflect the depth and value of this research.

7. The references, many of which are not in English, do not meet the requirements of this journal. Please revise them.

Reviewer 2 Report

Comments and Suggestions for Authors

See the comments in the attached document. I appreciate your work but the presentation is not clear enough.

For instance, you said you have only two samples. It is not enough for a conclusion.

I think the idea is interesting and useful in designing such structure, but the materials are not clearly described (I did not understand how is structured the composite with glass fibres (woven, unidirectional). In English, glue, adhesive and resin (as matrix) could be used more carefully. Also, I think bond would be joint and so on.

Comments on the Quality of English Language

To be improved with the help of a colleagure from the foreigh languages department.

Reviewer 3 Report

Comments and Suggestions for Authors

This manuscript by Ladnykh et. al. titled “ANN Prediction of Compliance Coefficients for Composite Mechanical Malleable Bonds of Timber Composite Elements” presents an interesting study on malleable bonds in Timber composite elements. A detailed study has been presented in this manuscript. However, a few points need attention and have been mentioned below:

1      The acronym used in the title “ANN” has not been defined anywhere in the text. Acronyms should be avoided in titles and every acronym used in the manuscript needs to be defined at the first place it is used.

2      Introduction: Most of the paragraphs have 2-3 sentences. Not sure why it was written this way. Authors need to reorganize this section.

3      For all the figures, it is suggested that these can be labelled/marked in the figures as it has been done for Figure 2. Figure like Figure 6 will especially need more labeling for the readers to understand the figure better. Otherwise figures like that of Figure 6, looks very jumbled and hard to interpret what is being represented.

4      Page 2, Line 58: Bullet points need to have sentence cases. Same for the other bullets used in this manuscript.

5      Page 4, Line 110: Last paragraph of the introduction needs to talk in details about what research gap is being focused with this work and how it has been solved by this manuscript. The authors need to have a good, detailed discussion about that in this last paragraph.

6      Page 4, Line 121, “In the 30s of 20th century…”: This paragraph needs to be a part of the Introduction section. The Materials and methods section should have all the details of the materials used and the methods followed in this work.

7       Page 7, Figure 8: The title of the figure is not self-explanatory. Authors need to modify it.

8      Page 10, Line 314, “Machine learning was performed…”: All the methods used need to be apart of the methods section and should not appear anywhere else in the manuscript.

9      Page 10, section 3.3, “The resuls of calculation”: Mistypes. Needs to be fixed. Similar errors need to be fixed.

   Page 11, Discussion: This part need to be rewritten. More explanation and detailed discussions are missing.

   Page 17, References: List of references are not written in the same format. It needs to be formatted. Along with it, it is suggested that the manuscripts written in English can be cited instead of those in any other language (Example: Ref #3, 6,7, 8, 9, 41 and others) so that readers globally can comprehend this manuscript.

Reviewer 4 Report

Comments and Suggestions for Authors

The work under review has many aspects that do not comply with the publication requirements, as follows:

1. First of all, it is not possible to identify the contribution of the authors because the research methodology is not explained.

2. The abbreviation ANN (ANN = artificial neural network) is used in the title, and the phrase machine learning algorithms is mostly used in the text of the article. Machine learning (ML) is used to teach machines how to handle the data more efficiently. Sometimes after viewing the data, we cannot interpret the extracted information from the data. In that case, we apply machine learning. So that the authors should explain what is the connection between these two processes and what is the contribution to the results of the research methodology used.

3. What is the reason for using the terminology "mechanical malleable bond" because it is not used in the specialized literature. Is it a name/definition/term proposed by the authors?

4. What did the authors want to say in the text "Wooden detail refers to a physically distinguishable component of a wooden element (Figure 2) [2]. For example, a wooden Derevyagin beam consists of two or more wooden details interconnected using oak dowels" ? It is erroneous to use the term "wooden detail" because it is known that "Derevyagin's beams are keyed beams precambered and connected with wooden plates" maybe made of oak.

5. Figures 1, 2 and 3 do not specify where those "malleable bonds" can be identified, and in figure 2 the specification with "timber details" is completely erroneous.

6. The phrase "It is described in the works [10–12] that through-sided and two-sided cracks appear in a wooden glued element during operation, as a result, the bearing capacity of a wooden element with such defects decreases and is equal to the bearing capacity of a composite element of equivalent cross-section" does not make sense - maybe the English translation is not correct.

7. The text from lines 69-81 seems like a compilation without logic and there are many parts in the same situation: 110-117 or 144-146 and many more.

8. Figure 5 is taken from reference 10 but without reference to the source.

9. There are many references to bibliographical sources in Russian that cannot be checked for similarities (figure 5 could be identified more easily).

10. In formula 1, the authors make a confusion between the bearing capacity and stresses. Bearing capacity is the maximum force and/or the maximum bending moment applicable to an element so that the element does not fail. The stresses that appear on the cross section of an element loaded with a force or a moment are the results distributed on cross-sections due the loading. In addition, as mentioned in the formula, there should have been + and - between the two terms that make up the member on the right, because the force can cause normal compressive or tensile stresses and the bending moment produces normal compressive and tensile stresses. Therefor when the force is compression the normal stresses are evaluated on the compression part of cross-section with + and with – for  stretched part.

11. In figures 7 and 8, the details necessary to be understood cannot be identified. In figure 8, the connection between the image and its explanation with "reinforcement coefficient" cannot be identified.

12. “maximum destructive load” must be replaced by “failure load”.

13. An explanation is needed for the reinforcement coefficient denoted by Karm.

14. An explanation is needed for the reinforcement coefficient denoted by Karm.

Chapter 4, Discussions, is not detailed enough. It contains conclusions and not what would lead to conclusions.

15. The Conclusions chapter contains explanations regarding the research methodology used that is not described in chapter 3, respectively the software program used.

16. The English translation has many mistakes. The terms used in English are from a direct translation (by Google translate) without using established terms in the specialized scientific literature. It is also observed in many situations the tiring and inappropriate repetition of some terms several times in a sentence.

Comments on the Quality of English Language

The English translation has many mistakes. The terms used in English are from a direct translation (by Google translate) without using established terms in the specialized scientific literature. It is also observed in many situations the tiring and inappropriate repetition of some terms several times in a sentence.